# Atomic-Approach to Predict the Energetically Favored Composition Region and to Characterize the Short-, Medium-, and Extended-Range Structures of the Ti-Nb-Al Ternary Metallic Glasses

**DOI:** 10.3390/ma12030432

**Published:** 2019-01-31

**Authors:** Bei Cai, Jianbo Liu, Jiahao Li, Menghao Yang, Baixin Liu

**Affiliations:** 1Key Laboratory of Advanced Materials (MOE), School of Materials science and Engineering, Tsinghua University, Beijing 100084, China; caib17@mails.tsinghua.edu.cn (B.C.); lijiahao@tsinghua.edu.cn (J.L.); mslbx@tsinghua.edu.cn (B.L.); 2Ames Laboratory, US Department of Energy, Ames, IA 50011, USA; myang@ameslab.gov

**Keywords:** Ti-Nb-Al system, interatomic potential, atomistic simulation, glass-formation ability (GFA), atomic structure

## Abstract

Ab initio calculations were conducted to assist the construction of the n-body potential of the Ti-Nb-Al ternary metal system. Applying the constructed Ti-Nb-Al interatomic potential, molecular dynamics and Monte Carlo simulations were performed to predict a quadrilateral composition region, within which metallic glass was energetically favored to be formed. In addition, the amorphous driving force of those predicted possible glassy alloys was derived and an optimized composition around Ti_15_Nb_45_Al_40_ was pinpointed, implying that this alloy was easier to be obtained. The atomic structure of Ti-Nb-Al metallic glasses was identified by short-, medium-, and extended-range analysis/calculations, and their hierarchical structures were responsible to the formation ability and unique properties in many aspects.

## 1. Introduction

Al-based metallic glasses were discovered around 1960, and they have plenty of potential applications owing to their unique properties, i.e., high corrosion resistance and excellent wear resistance, high strength, and superior hardness [1,2,3]. Moreover, the high-strength, but abnormally low-density, properties of Al-based amorphous alloys are of great significance for solving the energy-saving problems nowadays [4,5]. Al-based metallic glasses were first prepared in the Al-Si-TM (TM = Fe, Co, Ni, Cu) system [6,7]. The tensile strength of Al-Ln-TM amorphous alloys could be higher than 980 MPa [8], with the highest value of >1500 MPa in Al-Y-Ni-Co-Sc metallic glass [9]. Then the Vickers hardness and tensile fracture strength of Al_87_Y_8_Ni_5_ metallic glass could reach 300 DPN and 1140 MPa, respectively [8,10,11]. The recently discovered Ti-based metallic glass Ti_40_Zr_25_Ni_3_Cu_12_Be_20_ had superior plasticity and strength when deformed at a cryogenic temperature [12]. For Nb-based amorphous alloys, Nb-Si-X (X = V, Zr, Mo, Ta, W, C, B, and Ge) alloys have demonstrated a large amorphous-forming range, superconductivity, and remarkable mechanical properties [13]. Therefore, these experimental findings inspired the Ti-Nb-Al system to be selected for investigation. 

In practice, researchers frequently use some technical parameters to describe the glass-formation ability (GFA) of an alloy or to compare the GFAs of two alloys. However, the GFA of a metal system may vary while using different glass-producing techniques. Thus, many long-standing criteria for predicting the glass-forming ability (GFA) of alloys have been put forward [14,15,16], which are independent of the glass-producing techniques and facilitate the comparison of GFAs of different alloys, such as the reduced glass transition temperature *T*_rg_ (= *T*_g_/*T*_m_) [14], the supercooled liquid region Δ*T*_xg_ (= *T*_x_ − *T*_g_) [15], and the new parameter γ = *T*_x_/(*T*_g_ + *T*_m_) [16], where *T*_g_, *T*_x_ and *T*_m_ are the glass transition temperature, onset crystallization temperature, and melting temperature, respectively. However, these criteria are still empirical or semi-empirical regulations, sometimes limited in predicting the GFA of metallic glasses and failing to provide valuable guidance for the composition design and preparation of metallic glasses [17,18]. Hence the development of glassy alloys urgently requires exploration of more effective theoretical methods to predict the GFA and glass-formation range/region (GFR). Long-standing research has shown that metallic glass formation is an intrinsic property of the alloy system, and the interactions between atoms determine the physicochemical properties [19]. Therefore, it is possible to predict GFAs of glassy alloys from atomic-scale, drawing supports from molecular dynamics (MD) and Monte Carlo (MC) simulations by constructing the inter-atomic interaction potential [20].

Apart from the formation mechanism of metallic glass, the atomic structure of metallic glass is also one of the most challenging fundamental problems in this field. It is necessary to mention that different atomistic simulation procedures, such as liquid melt quenching (LMQ) and solid-state amorphization (SSA), yield similar atomic structures [21]. The physical description and modeling of the atomic structure will help in essentially revealing the structural origin of metallic glass formation and properties [22,23]. Owing to the close packing in space and energy minimization, the atomic packing mode possesses short-range order (SRO) and medium-range order (MRO), but expresses long-range disorder. On the SRO scale, we mainly analyzed the local neighboring clusters around atoms [24,25,26,27,28,29,30,31]. It has been authenticated by experimental studies that icosahedral or icosahedral-like clusters are the basic building component in metallic glasses. The interconnection of local clusters further form MRO [32,33,34,35,36,37,38]. The characteristics of MRO shed new light on understanding of the mechanical properties and stability of monolithic metallic glasses. Sheng et al. proposed the quasi-cluster packing model [33], in which clusters establish an interpenetrated network through the vertex, edge, face, and volume linkages [33,35,39], and show icosahedral-like packing on the MRO scale. Compared with other types of linking modes, the icosahedral network is structurally more stable and more energy efficient. Ichitsubo et al. [40] observed the existence of so-called “hard zones” and “soft zones” in Pd-Ni-Cu-P metallic glass. The heterogeneity of structure and properties in metallic glass can be traced back to the heterogeneous connectivity among different cluster-types [41,42,43,44]. The interpenetrating linkages of icosahedra-like and icosahedra exhibit strong correlation in space and tend to aggregate and form an even higher hierarchical level, i.e., structural skeleton. Therefore, clarifying the multi-hierarchical structural characteristics is vital to further demonstrate the structure-property relationship in metallic glass.

## 2. Construction of Ti-Nb-Al Interatomic Potential

A long-range empirical potential [45,46,47] was constructed for the Ti-Nb-Al system. The total energy *E_i_* of atom *i* can be computed by:(1)$$E_{i}=\frac{1}{2}\sum_{j\neq i}V(r_{ij})-\sqrt{\sum_{j\neq i}\Phi (r_{ij})}$$

Here *r_ij_* is a distance between atoms i and j, and the two terms, *V*(*r_ij_*) and Φ(*r_ij_*) of Equation (1), are the pair part and the electron density term, respectively. They can be expressed as:
(2)$$V(r_{ij})={\left(r_{c1}-r_{ij}\right)}^{m}(c_{0}+c_{1}r_{ij}+c_{2}r_{ij}{}^{2}+c_{3}r_{ij}{}^{3}+c_{4}r_{ij}{}^{4}),\;0<r_{ij}\leq r_{c1}$$
(3)$$\Phi (r_{ij})=\alpha^{2}{\left(r_{c2}-r_{ij}\right)}^{n},\;0<r_{ij}\leq r_{c2}$$

Here, *r_c_*_1_ and *r_c_*_2_ are designed for the pair part and electron density term, respectively. *m* and *n* are integers adjusted according to the specific metals, and *α* and *c_i_* (*i* = 0,1,2,3,4) are the potential parameters to be determined through the fitting process.

In this work, we used a MATLAB code developed by the researcher’s group for the fitting of the potential parameters. The Ti-Nb-Al system includes six sets of potential parameters for Ti-Ti, Nb-Nb, Al-Al, Ti-Nb, Ti-Al, and Nb-Al, of which the Nb-Nb, Al-Al, and Nb-Al cross-potential parameters have been previously fitted by the author [48]. The physical properties (the lattice constants, elastic modulus, elastic constants, and cohesive energies) of hcp-Ti metal were obtained from the relevant literature [49], while the physical properties of the metastable bcc- and fcc-Ti were calculated by ab initio, their physical properties then applied to fit potential parameters. When fitting the cross-potential parameters of Ti-Nb and Ti-Al, several intermetallic compounds with different structures and alloy compositions were selected to cover as wide a composition range as possible in the binary alloy phase diagrams. Ab initio calculations based on density functional theory (DFT) aided the construction of n-body potential, which was conducted using CASTEP in Material Studio [50,51,52] in order to ensure that the calculation met the energy convergence and the accuracy requirements. In the calculation process, the exchange correlation functional was constructed based on the generalized gradient approximation of Perdew and Wang (PW91) [52]. The Brillouin zones of different structures employed the same k-points grid separation of 0.02 Å^−1^. The cutoff energy of the plane wave was 700 eV.

Table 1 lists the potential parameters of Ti-Nb-Al ternary alloys. To verify whether the potential parameters obtained in this work were reasonable, Table 2, Table 3 and Table 4 show the physical properties derived from n-body potential. They all matched well with the experimental or ab initio calculated values, indicating that the constructed potential could reasonably describe the static physical properties of intermetallics.

We further examined whether the potential was consistent with the atomic interactions in a non-equilibrium state, i.e., comparing the equation of state (EOS) derived from the constructed potential with the Rose Equation [53]. As shown in Figure 1, the EOS curve was basically consistent with the ROSE curve, especially near the equilibrium position *a* = *a*_0_. Hence, it could be further applied to MD simulations to explore the scientific issues even farther away from the equilibrium state, i.e., the glass formation mechanism and its atomic-scale structure.

## 3. Metallic Glass Formation of Ti-Nb-Al System

### 3.1. Atomic Simulation Methods

The preparation of metallic glass is always far away from the equilibrium state, and the dynamic conditions are very strict and harsh. Therefore, the rearrangement and diffusion of atoms in the alloy are greatly hindered, leading to the difficulty of nucleation and growth of those intermetallics with complex structures. At this point, the competing phase against amorphous is the solid solution with a simple structure, such as the hcp, bcc, and fcc structure [54]. Consequently, the problem of predicting the GFR of a Ti-Nb-Al alloy system could be converted into one of comparing the relative stability of the solid solution phase with the metallic glass within the full composition triangle. It would then be possible to determine the composition region in which the metallic glass presented as stable.

From a physics perspective, based on the constructed atomic interaction potentials, at least 231 initial solid solution models were established within the entire composition triangle of the Ti_x_Nb_y_Al_1−x−y_ system, given the composition interval of x, y = 5 at.%. It is possible to predict the GFR of the alloy system with the help of Large-scale Atomic/Molecular Massively Parallel Simulator (LAMMPS) packages [55,56]. For the Ti-Nb-Al ternary system, we achieved the target composition of the constructed model by replacing solvent atoms with solute atoms at random. When the dominant component of the alloy composition is Ti, the hcp initial solid solution model is constructed. Similarly, for the Nb-based and Al-based alloys, we applied the bcc and fcc initial solid solution models, respectively. For the hcp model, the (100), (120) and (001) crystalline directions are designed to be parallel to the x-, y- and z-axes. Whereas for both bcc and fcc models, the x-, y-, and z-axes are parallel to three crystalline orientations (100), (010), and (001), respectively. Periodic boundary conditions were applied to the x-, y-, and z-coordinate directions. Considering the simulation time on the atomic scale, the size of the simulated cells was set to consist of 4000 (10 × 10 × 10 × 4) atoms, 3456 (12 × 12 × 12 × 2) atoms, and 6912 (12 × 12 × 12 × 4) atoms for the hcp, bcc, and fcc solid solution models, respectively. MD and MC simulations were adopted in this work. For MD simulations, the isothermal-isobaric (NPT) ensemble with a time step of 5 femtoseconds was selected [57], and the simulation temperature and pressure were designed by the Nose–Hoover thermostat and barostat to be 300 K and 0 Pa, respectively [58,59]. Moreover, the simulation process ran for 1 × 10^6^ MD time steps to reach a stable state. MC simulations were proceeded for thousands of steps at 300 K, 0 Pa under NPT ensemble to fully relax the initial solid solution model. Atom displacement and box deformation are the two types of “moves” during MC simulations. From a thermodynamics viewpoint, the energy difference between the amorphous phase and the real solid solution, is the driving force for the solid solution to amorphize.

The structural transition from crystal to amorphous is monitored by the pair correlation function [60] *g*(*r*), which can clearly describe the distribution of bond lengths between atoms and the disordered trend of structures with increasing distance. Details for *g*(*r*) equation can be found in Ref. [60].

According to the Voronoi tessellation analysis [61,62,63], an envelope of a family of perpendicular bisectors between a central atom and all of its nearby atoms constitutes the surface of a Voronoi polyhedron for the central atom, which can be differentiated by the indices <*n*_3_, *n*_4_, *n*_5_, *n*_6_,…>, where *n_i_* stands for the number of *i*-edged faces of the Voronoi polyhedron and is the total coordination numbers (CNs). In addition, 3D atomic configurations were applied to assist in the analysis of the alloy structure. In this article, we mainly focused on analyzing and characterizing microscopic heterogeneity and multiple hierarchical structure under the optimal composition of Ti_15_Nb_45_Al_40_. Finally, the formation mechanism and structure of metallic glass are linked.

### 3.2. Glass Formation Region of Ti-Nb-Al System

We now discuss the structure factor *g*(*r*) and the 3D atomic position projection for two alloys in the Nb-rich corner, i.e., Ti_5_Nb_90_Al_5_ and Ti_10_Nb_70_Al_20_. Figure 2a,b illustrated that the atomic configuration of Ti_5_Nb_90_Al_5_ alloy was similar to that of bcc structure and remained in a crystalline state after relaxation. When the Ti content increased to 10 at.%, it could be seen from Figure 2c,d that the atomic arrangement was disordered, and the Ti_10_Nb_70_Al_20_ alloy underwent a crystal-to-amorphous transition, resulting in the formation of metallic glass. It revealed that the underlying physical process of the metallic glass formation was the collapse of the crystalline lattice while the solute concentrations exceeded the critical solid solubility.

Figure 3 explained the GFR of Ti-Nb-Al alloy system. The gray solid circles and hollow circles represented the alloy composition of the formed amorphous and crystal, respectively. When the alloy composition fell in the central quadrangle region surrounded by ABCD, the lattice of the solid solution rapidly collapsed, leading to the formation of metallic glass, while it was a stable crystalline region outside the ABCD quadrilateral. To check whether the predicted GFR was reliable, the composition points of Ti-Nb-Al metallic glass that were prepared experimentally were compared with the predicted range, as shown in Figure 3 with colored symbols. In the Al-Ti system, the amorphous phase was prepared by ion beam mixing (IBM) [64], and the preferred composition range was found to be at least 25–75 at.% Ti. Experimentally, Al-Ti bulk metallic glasses were prepared by liquid melt quenching (LMQ) [65]. The synthesis of amorphous powder was also attained by mechanical alloying (MA) for the composition range Ti_x_Al_1−x_ (50 < x < 80 at.%) [66,67]. In addition, Al_x_Nb_1−x_ (15 < x < 75 at.%) alloys were prepared by MA of elemental crystalline powders [68]. Powders of Ti, Nb, and Al were raw materials, Ti_52−x_Nb_48_Al_x_ (x = 2, 4, 6, 8 at.%), Ti_65_Nb_11_Al_24_, Ti_50_Nb_25_Al_2_5, Ti_50_Nb_37.5_Al_12.5_, and Ti_47.6_Nb_28.5_Al_23.9_ amorphous alloys were mechanically alloyed [69,70]. It could be seen from Figure 3 that the amorphous components obtained by experiments all fell within the blue quadrilateral ABCD, which proved that the GFR of Ti-Nb-Al metallic glass predicted by MD simulations was reliable and reasonable.

### 3.3. Optimization of Glass—Formation Compositions

Next, we studied the composition point of the Ti-Nb-Al system that had the largest glass-forming ability. The energy difference between solid solution and amorphous is defined as the amorphous driving force (ADF). As the energy difference increases, the ADF and corresponding GFA possessed by alloy are both larger, and the metallic glass is more and more easily to be prepared. Figure 4 showed the ADF of each composition point calculated in Ti-Nb-Al system. From the energy scale on the right side, it was known that the energy difference between the amorphous phase and the solid solution phase was negative, i.e., the metallic glass was easily formed within the predicted GFR. The composition point Ti_15_Nb_45_Al_40_ with the black pentagram symbol possessed the largest GFA, and metallic glass was most likely to be formed in the pentagram and its vicinity. To verify whether the GFA calculated was reasonable or not, the predictions were further compared with the experimental values marked in Figure 3. It could be seen that the composition points of the metallic glass that were experimentally prepared were basically concentrated in the optimized composition region near Ti_15_Nb_45_Al_40_. The simulated GFAs also agreed with the experimental data.

## 4. Atomic Structure of Ti-Nb-Al Metallic Glass

### 4.1. Short-Range Order and Chemical Microscopic Heterogeneity in Metallic Glass

The Voronoi polyhedron method was employed to analyze the SRO of metallic glass. As shown in Figure 5, the distribution of eight dominant Voronoi polyhedrons in Ti_15_Nb_45_Al_40_ alloy was calculated by Voronoi tessellation methods. It could be seen from Figure 5a that when Ti was used as a core, the polyhedral index with the highest content was <0,3,6,4>, followed by <0,3,6,3> and <0,2,8,2> clusters, while the atomic configuration of <0,3,6,4> cluster was shown as the embedded graph. All the three clusters defined as icosahedral-like had high local quintic symmetry and could be considered as an ideal icosahedral polyhedron that has undergone a distortion [34]. The polyhedrons with quartic or sextic symmetry assisted in the efficient packing of clusters in metallic glass [71]. It can be seen from Figure 5b that, when it was Nb-centered, the clusters with the highest proportion of the top three were <0,2,8,4>, <0,3,6,5>, and <0,1,10,2> clusters, respectively. As to Al-centered, a similar conclusion could be drawn from Figure 5c. These clusters could serve as basic building blocks of Ti_15_Nb_45_Al_40_, of which most were Kasper-type polyhedrons. Furthermore, they could be aggregated into MRO by inter-cross-sharing (IS) linkages.

Figure 6a–c analyzed the distributions of total and partial CN in (TiNb_3_)_1−x_Al_x_ (x = 80, 60, 40 at.%) metallic glasses. It could be seen initially the polyhedrons with CN = 11, 12 and 13 possessed the highest proportion. Further inspecting the Figure, one can see that the dominant total coordination number was CN = 11 and 12. When it was Ti-centered, the Ti concentration was low at the beginning, and the trend of the curve changed slowly. As the Al concentration decreased from 60 to 40 at.%, the dominant CN of Ti (the red curve) moved to the right from CN = 11 to CN = 12. When it was Nb-centered, as the Al concentration decreased from 80 to 60, 40 at.%, the dominant CN of Nb (the green curve) moved to the right from CN = 11 to CN = 12, 13 and then to CN = 13, 14. Whereas the dominant clusters of Al-centered were similar to that of the total, the fraction of CN = 11 and CN = 12 account for the most. Because Nb possessed smaller atomic radii than Ti and Al, elements with smaller atomic radii required fewer short-range local packing unit to form clusters. Consequently, most of the coordination polyhedrons with CN > 12 were Nb-centered. Figure 6d–f listed the distributions of the dominant coordination polyhedrons centered on each unitized element to further deduce the atomic configurations. The main coordination polyhedron in Ti_5_Nb_15_Al_80_ was shown in Figure 6d; the concentration of Al-centered clusters was larger than that of Ti- and Nb-centered, the fractions of <0,2,6,3>, <0,3,6,4>, and <0,2,6,4> clusters account for the most. For the coordination polyhedron of Ti_10_Nb_30_Al_60_ metallic glass shown in Figure 6e, it could be seen that the prevailing clusters were indexed as <0,2,8,2>, <0,2,6,4>, <0,2,6,3>, and <0,3,6,4>, respectively. As the Al concentration decreased to 40 at.% shown in Figure 6f, the Al-centered <0,3,6,3> cluster possessed the largest fraction. It could be seen from Figure 6d–f that, as the Al concentration decreased, the fractions of <0,3,6,3> coordination polyhedrons increased correspondingly. However, the fraction of <0,3,6,4> and <0,2,6,3> coordination polyhedrons decreased. Overlooking the distribution of Voronoi polyhedrons in (TiNb_3_)_1–x_Al_x_ (x = 80, 60, 40 at.%) metallic glass, we found that the preponderant interconnected icosahedral-like clusters were <0,3,6,3>, <0,2,8,1>, <0,2,8,4>, <0,2,8,5>, and <0,2,8,2>. It was known that icosahedral and icosahedral-like polyhedrons have the effect of stabilizing the structure of metallic glass because of energy minimization and efficient atomic packing [72].

In the previous section, the topological SRO in Ti_15_Nb_45_Al_40_ metallic glass was analyzed. Next, we took Ti_15_Nb_45_Al_40_ metallic glass as an example to further analyze the relationship of chemical SRO between crystalline phases and amorphous. The chemical SRO essentially described the extent to which the local atomic type deviated from the integral composition and characterized the degree of chemical microcosmic inhomogeneity presented in metallic glass. The CN distribution of the nearest neighbor atoms around Ti, Nb, and Al was shown in Figure 7, where the color depth of the grid indicated the percent value, referring to the color scale on the right for details. The double-index (*m*, *n*) represented the local chemical environment of each atom, where *m* denoted the number of Al neighbors, *n* represented the Ti and Nb neighbors, and Ti and Nb were not distinguished, for the time being. It was assumed here that Ti and Nb were quasi-neighbors to obtain a neat display. *m* + *n* was equal to the total CN. It could be seen from Figure 7 that the (*m*, *n*) indices around Ti, Nb and Al atoms regularly fell into a striped area, which was bordered by two boundary lines of *m* + *n* = CN*_i_* (*i* = min, max). The white dotted line (4:6) represented the corresponding apparent stoichiometry line. It could be seen that the local chemical environment in the metallic glass was dispersed and largely deviated from the apparent alloy composition, demonstrating the existence of chemical SRO. Further analysis of the double-index (*m*, *n*) with the highest percentage of Ti-, Nb-, and Al-centered revealed that the Ti/Nb atoms were more prone to neighboring Al atoms, that is, it contained more Al atoms compared with the apparent chemical composition. This was consistent with experimentally observed negative mixing enthalpy of Ti-Al and Nb-Al in Table 5, which facilitated mutual attraction among the corresponding atoms to meet the energy minimization.

In general, the topological and chemical SRO of metallic glass is more complex than that of crystalline phase. It promotes the crystallization resistance of the metallic glass but is not conducive to the nucleation of the intermetallic compound, thereby increasing the stability of the amorphous phase. In the meantime, the chemical SRO leads to a variety of cluster-types in metallic glass, increasing the efficient packing in space to reach structural stability and energy minimization [74].

### 4.2. Medium-Range Order, Cluster Correlation Heterogeneity and Atomic Volume

On the MRO scale, local clusters overlap and interconnect with each other by sharing vertex, edge, face, or volume, i.e., VS, ES, FS, and TS. With the effective packing mode, the structural skeleton and percolated network of metallic glass is finally formed. The connection among local clusters is often tendentious and in-homogeneous, which directly leads to structural heterogeneity in metallic glass. For example, it explains the mechanical heterogeneity in the deformation process. Therefore, establishing an effective method to characterize correlation heterogeneity among enormous number of cluster-types is very crucial for further exploration of the structure-property relationship.

In this paper, we proposed a cluster connection factor (CCF) [75] to distinguish between the neighboring preferences of different cluster-types. The expression is defined as follows:
(4)$$F_{\mathrm{mn}}=\frac{C_{\mathrm{mn}}}{C_{m}N_{n}}N-1.$$

Here, *F_mn_* ≠ *F_nm_*, *C_mn_* represents the number of linkages between two *m*- and *n*-type clusters. Note, only the case of connections via TS linkages will be considered here. *C_m_* is the total number of linkages between the *m*-type cluster and all the other types of clusters, *N_n_* is the number of n-type clusters in the alloy, and *N* represents the total number of clusters of all types. If *F_mn_* < 0, a repulsive-like interaction appears between *m*- and *n*-type clusters. If *F_mn_* > 0, *m*- and *n*-type clusters tend to inter-linkage. If *F_mn_* = 0, a random packing behavior occurs among different cluster-types.

For the selected eight typical Voronoi clusters of Ti_15_Nb_45_Al_40_ metallic glass, the calculated matrix of CCFs was shown in Figure 8. The color scale on the right indicated the corresponding CCF value. It could be seen from Figure 8 that there was a very strong connection heterogeneity on the MRO scale, and the packing mode diverged remarkably from the random packing mode. The Ti <0,3,6,3> and Ti <0,3,6,4> clusters exhibited intensive aggregation and interconnected tendencies. In terms of horizontal and vertical integration, Nb <0,2,8,4> were Kasper-type clusters with strong local quintic symmetry. It showed a favorable tendency to interconnect with other types of clusters, so it could serve as a hinge for interconnecting clusters. Due to the fact that the Nb element possessed the highest content in Ti_15_Nb_45_Al_40_ metallic glass, it could be treated as solvent atoms to promote the inter-connectivity of clusters centered on Ti and Al solute atoms. Meanwhile, Nb-centered clusters and Ti-, Al-centered clusters also aggregated and cooperated with each other, which was beneficial to increase the packing efficiency in space and thus achieve the purpose of strengthening the structural stability of glassy alloys.

Zhao and Louzguine–Luzgin et al. have investigated the structural relaxation and its influence on the mechanical properties and elastic properties of Cu-Zr-Ti-Pd and Mg-Zn-Ca bulk metallic glass (BMG) [76,77]. It was found that the distribution of free volume has an effect on the elastic properties of metallic glass. During the stage of plasticity deformation in glassy alloys, the macroscopic strengthening effect is manifested by the atomic micro-strengthening in shear band. The free volume (FV) is confined to the cage formed by “backbone”. Through the analysis of the atomic volume, with the increase of deformation, the growth and aggregation of FV become more and more difficult, the “backbone” becomes more and more dense, and eventually the deformation resistance is increased. Therefore, the study of atomic volume is of great significance. It preliminarily establishes the relationship between the amorphous structure and their macroscopic mechanical property. In this paper, the Voronoi tessellation of the simulation cell was calculated by the open visualization tool (OVITO) [78], with the particle positions as Voronoi polyhedron centers. By default, the volume of the Voronoi polyhedron are output by the analysis modifier, and the spatial bin is aligned parallel to the simulation box x-, y-, and z-axes. The atomic Voronoi volume calculated by proper averaging procedure can accurately measure the local density. In a given region, the larger Voronoi volume corresponds to the larger local FV, which involves the smaller local density in the region. Figure 9 showed the atomic volume distribution of Ti_15_Nb_45_Al_40_ metallic glass on the X–Y, X–Z and Y–Z plane of projection. As could be seen from Figure 9, along the x- and y-axis, the size and distribution of atomic volume in the Ti_15_Nb_45_Al_40_ metallic glass was greatly similar, and both were larger than the atomic volume along the z-axis, indicating that there were more FV along the x- and y-axis than that along the z-axis. Figure 10 exhibited the average atomic Voronoi volumes along the z-axis of the (TiNb_3_)_1−x_Al_x_ (x = 5–90 at.%) system. It could be seen that as Al concentration increased, the average atomic volume decreased firstly and then increased. Its value was minimum at Al = 40 at.%, i.e., the Ti_15_Nb_45_Al_40_ glassy alloy possessed the smallest average atomic volume, the highest packing efficiency, and the lowest alloy energy.

### 4.3. Structural Skeleton in the Extended Scale

It could be known from Figure 8 that Nb <0,2,8,4> served as a hinge for interconnecting clusters. Hence, we took the Nb <0,2,8,4> cluster as the starting point of analysis. With the help of various structural characterization methods, the evolution of the glassy alloy from local cluster on the SRO scale and quintic-like packing on the MRO scale, to structural skeleton on an expanded scale was shown in Figure 11. The Nb <0,2,8,4> clusters on the SRO scale mainly focused on the configuration of the nearest neighbor atoms. On the MRO scale, Nb-centered icosahedral-like clusters could interconnect with about fourteen solute atoms-centered [25] clusters. They interconnect with each other by VS, ES, FS, and TS linkage-types. Ultimately, a nano-scale super-cluster was engendered by the effective packing of neighboring clusters. For the connection mode of the structural skeleton, the extracted cross-linked patch was shown in Figure 11, which contained 40 clusters. It could be seen that each cluster in the presented patch mainly interpenetrated with each other to form a cross-linked network in the form of TS linkages [39]. However, the VS, ES, and FS linkages assisted in constructing the network and helped to strengthen the stability of the metallic glass and affect the macroscopic properties of material [79].

## 5. Further Discussion about the GFA and Atomic Structure

Combined with the analysis in Figure 2 above, it could be found that the dominant interconnected Voronoi polyhedra in Ti_15_Nb_45_Al_40_ glassy alloy were <0,2,8,5>, <0,2,8,4>, and <0,3,6,4>. Adding the most stable icosahedral cluster <0,0,12,0>, Figure 12 illustrates the relationship between the above dominant interconnected clusters of the (TiNb_3_)_1−x_Al_x_ system and the Al concentration. It was analyzed that the population of the four dominant clusters initially increased as Al concentration increased, until it reached the maximum value when the Al concentration was 15 at.%. It was followed by the reduced content as further increasing Al concentration. The aforementioned trend was consistent with the GFA of (TiNb_3_)_1−x_Al_x_ metallic glass. It was further illustrated that there was a great difference in the topological SRO between the metallic glass and its corresponding crystal. The topological SRO would increase the crystallization resistance of metallic glass and consolidate the stability, thereby explicating the structural causation of the GFA. Therefore, the GFA in the Ti-Nb-Al system would mainly come from the instability of the crystalline phase. GFA determined from both structural and energetic perspectives showed a good consistency.

## 6. Concluding Remarks

Based on the constructed Ti-Nb-Al interatomic potential, MD and MC simulations not only clarified the mechanism of the metallic glass formation, but also predicted an energetically favored quadrangular metallic glass-formation region. Further calculations of the amorphous driving force for all of the possible glassy alloys in the predicted region pinpointed an optimized composition at Ti_15_Nb_45_Al_40_, and it could correspond to the one with a largest glass forming ability in the Ti-Nb-Al ternary alloy system. By means of multiple structural characterizations, a series of hierarchical structural analysis to the SRO, MRO, and even an expanded scale was performed for the Ti_15_Nb_45_Al_40_ metallic glass. The results showed that a large degree of topological and chemical SRO existed inside the Ti_15_Nb_45_Al_40_ glassy alloy, and in the MRO scale, with the aid of the CCF, it was revealed that a significant structural heterogeneity existed. By observing the variation of atomic volume, it was concluded that the Ti_15_Nb_45_Al_40_ metallic glass could possess the highest packing efficiency and lowest alloy energy. The changes in the content of the dominant clusters were consistent with the GFA of (TiNb_3_)_1−x_Al_x_ alloys. In addition, in the structural evolution from SRO and MRO to the structural skeleton, we explored the structural origin of the glass-forming ability, as well as the unique properties of the Ti-Nb-Al metallic glasses.

## Figures and Tables

**Figure 1 materials-12-00432-f001:**
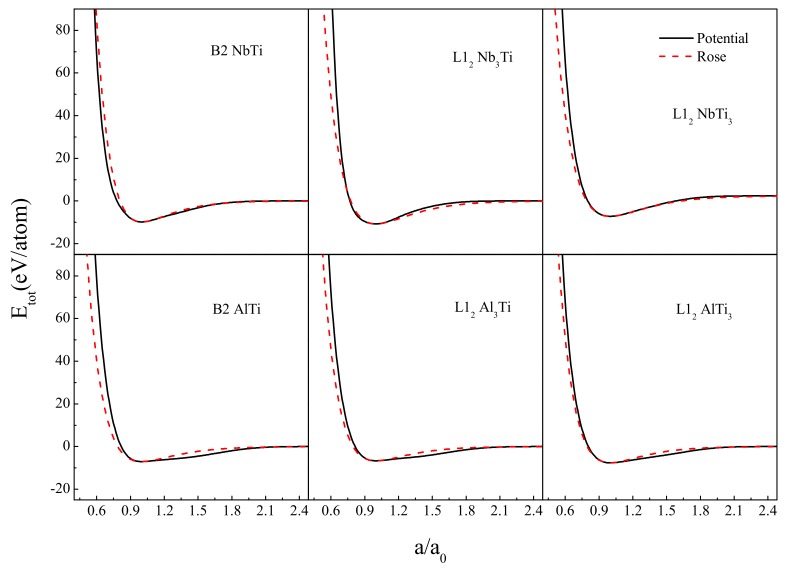
Potential energies (solid line) as a function of lattice constant compared with the Rose equation (dash line).

**Figure 2 materials-12-00432-f002:**
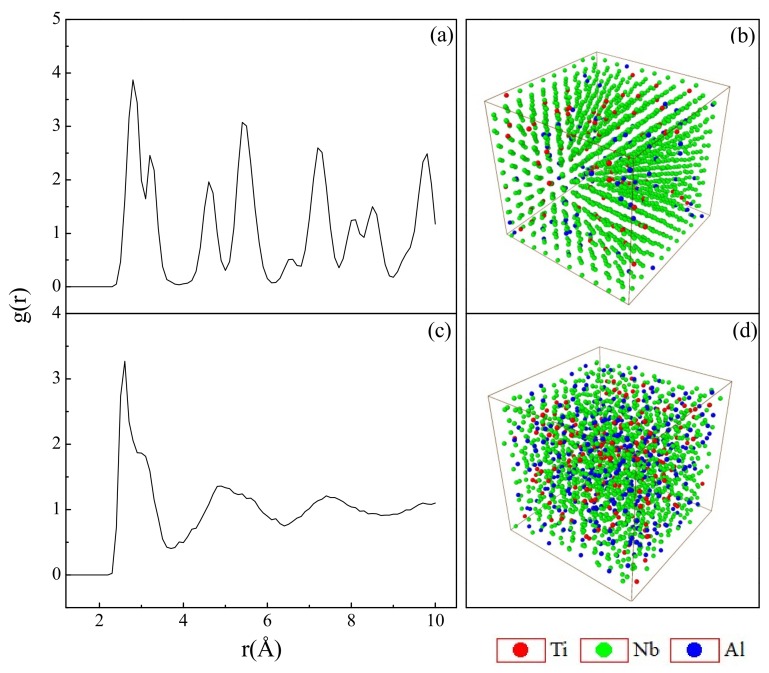
The *g*(*r*) and 3D atomic position projections for (**a**,**b**) Ti_5_Nb_90_Al_5_ (crystalline state) and (**c**,**d**) Ti_10_Nb_70_Al_20_ (amorphous state), respectively.

**Figure 3 materials-12-00432-f003:**
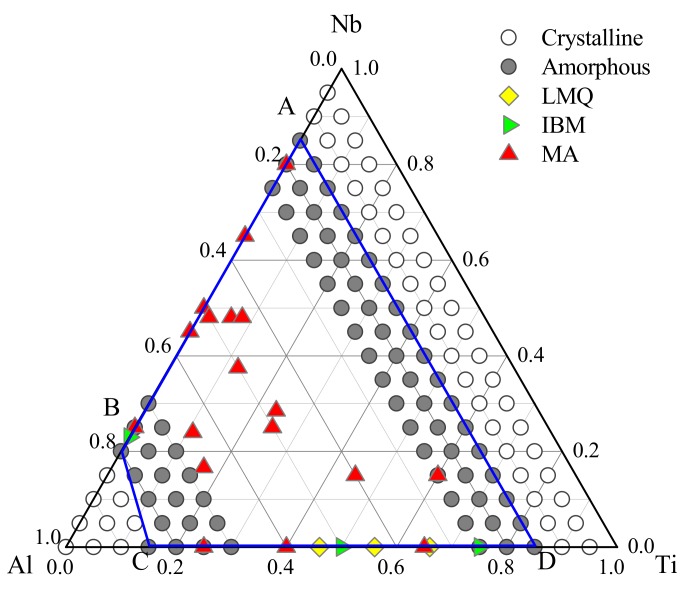
The glass formation range/region (GFR) (inside the quadrilateral) of the Ti-Nb-Al system obtained from Molecular Dynamics (MD) simulations.

**Figure 4 materials-12-00432-f004:**
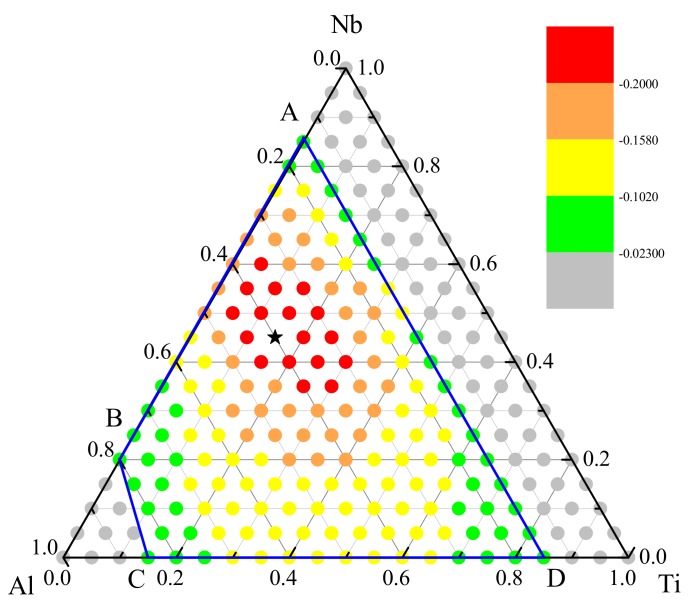
The amorphous driving force (eV/atom) of Ti-Nb-Al solid solutions in the predicted GFR. The black pentagram indicates the alloy with the largest glass-formation ability (GFA).

**Figure 5 materials-12-00432-f005:**
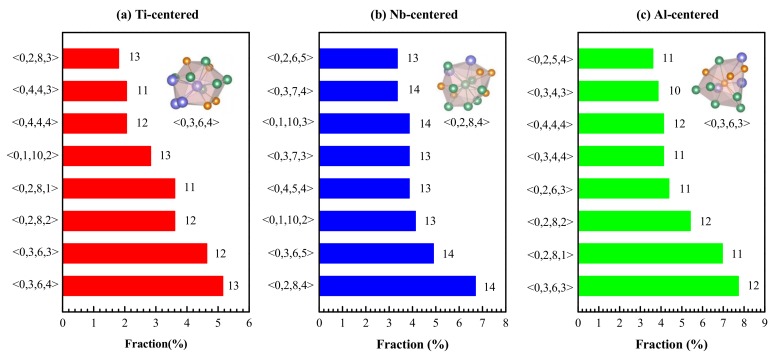
The distribution of Voronoi polyhedrons around Ti, Nb, and Al atoms in Ti_15_Nb_45_Al_40_ glassy alloy. (**a**) Ti-centered; (**b**) Nb-centered; (**c**) Al-centered.

**Figure 6 materials-12-00432-f006:**
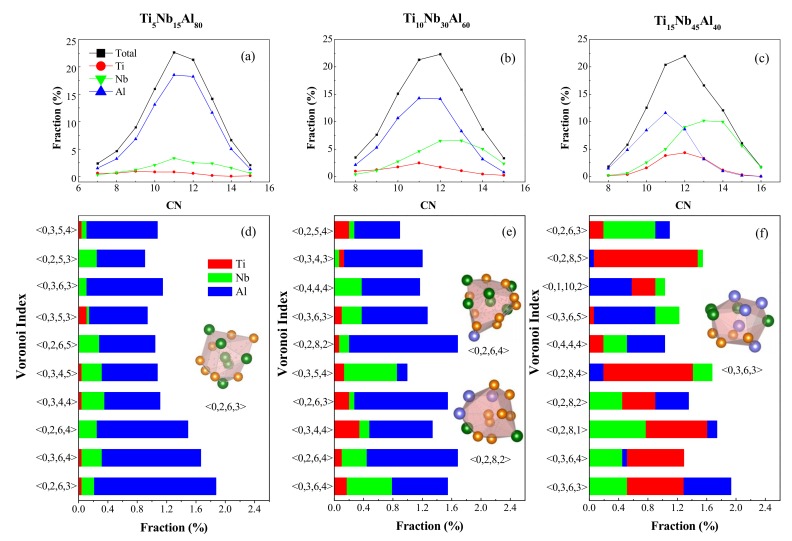
The distribution of Voronoi polyhedrons around Ti, Nb, and Al atoms in Ti_15_Nb_45_Al_40_ glassy alloy. (**a**–**c**) The distributions of total and partial CN in (TiNb_3_)_1−x_Al_x_ (x = 80, 60, 40 at.%) metallic glasses. (**d**–**f**) The distributions of the dominant coordination polyhedrons centered on each unitized element.

**Figure 7 materials-12-00432-f007:**
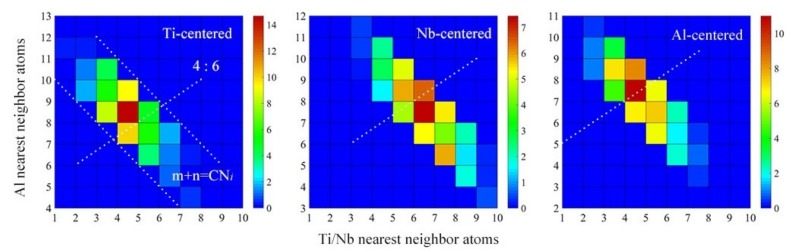
The distribution of the nearest neighboring atoms around Ti, Nb, and Al in Ti_15_Nb_45_Al_40_ glassy alloy.

**Figure 8 materials-12-00432-f008:**
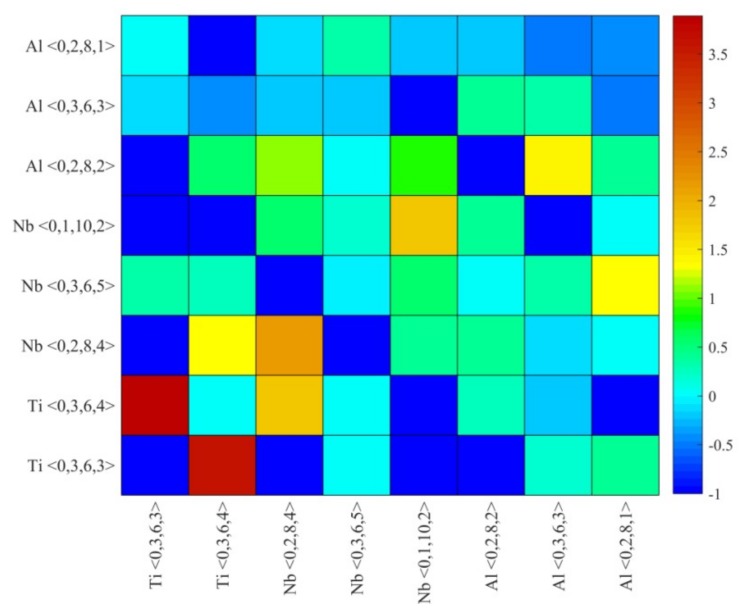
The distribution of connection factor *F_mn_* among eight typical Voronoi clusters in Ti_15_Nb_45_Al_40_ glassy alloy.

**Figure 9 materials-12-00432-f009:**
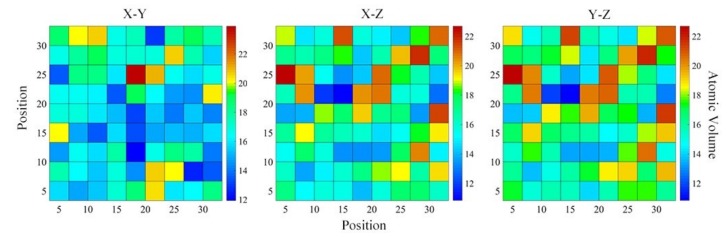
The distribution of atomic volumes along z-, y-, and x-axis in Ti_15_Nb_45_Al_40_ glassy alloys, respectively.

**Figure 10 materials-12-00432-f010:**
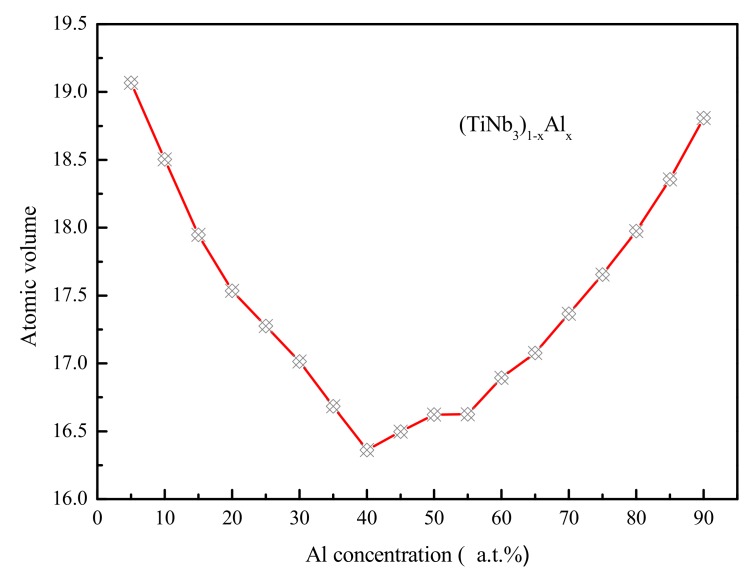
The variation of atomic volume as Al concentration increasing in (TiNb_3_)_1−x_Al_x_ alloys.

**Figure 11 materials-12-00432-f011:**
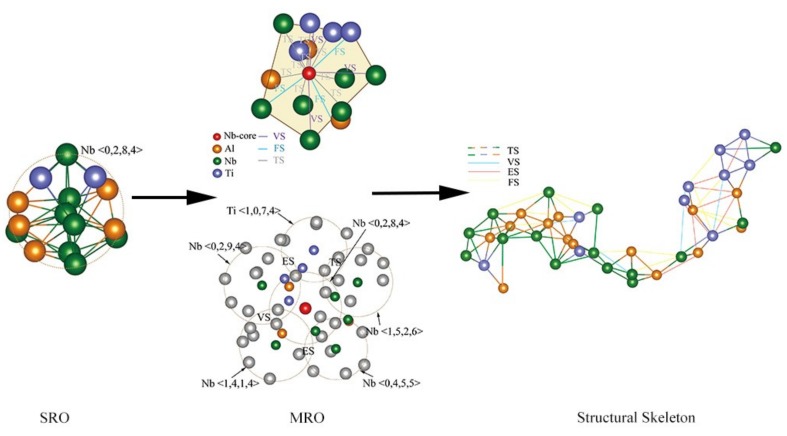
The variation of atomic volume as Al concentration increasing in (TiNb_3_)_1−x_Al_x_ alloys.

**Figure 12 materials-12-00432-f012:**
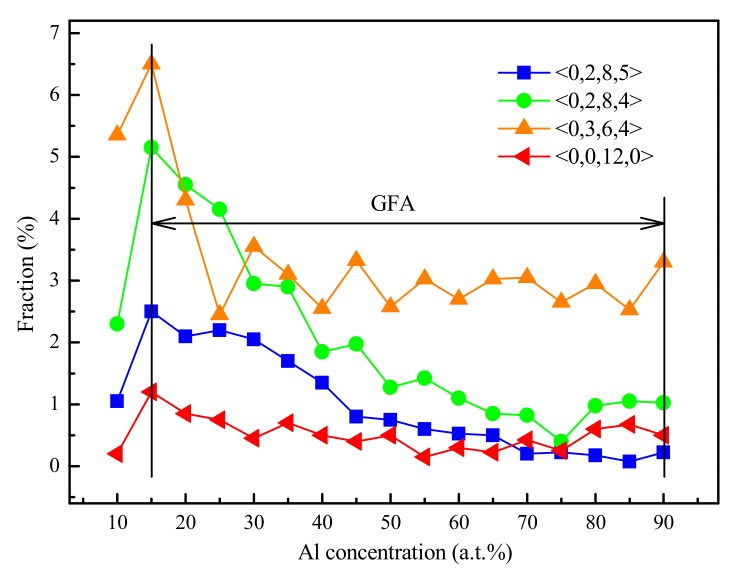
The distribution of main Voronoi clusters in (TiNb_3_)_1−x_Al_x_ glassy alloys.

**Table 1 materials-12-00432-t001:** The potential parameters for the Ti-Nb-Al system.

Potential Parameters	Ti-Ti	Nb-Nb	Al-Al	Ti-Al	Nb-Al	Ti-Nb
*c*_0_ (10^−19^ J/Å*^m^*)	1.146137	16.721753	0.558355	0.309051	1.720707	5.543857
*c*_1_ (10^−19^ J/Å^*m* + 1^)	−1.417844	−22.891094	−0.621271	−0.338078	−2.09683	−8.078405
*c*_2_ (10^−19^ J/Å^*m* + 2^)	0.700512	11.808389	0.259702	0.140656	0.951759	4.703939
*c*_3_ (10^−19^ J/Å^*m* + 3^)	−0.159982	−2.695603	−0.047246	−0.025314	−0.188004	−1.241254
*c*_4_ (10^−19^ J/Å^*m* + 4^)	0.014262	0.228706	0.003148	0.001621	0.013434	0.124878
*α* (10^−19^ J/Å*^n^*)	0.056220	−0.02174	0.035311	0.062566	0.072146	0.108197
*m*	4	4	4	4	4	4
*n*	6	8	6	6	6	6
*r_c_*_1_ (Å)	5.171296	4.802774	5.826946	6.551139	5.525992	4.499325
*r_c_*_2_ (Å)	6.935479	6.818573	7.246908	7.068235	6.452469	6.567841

**Table 2 materials-12-00432-t002:** The lattice constants (*a*), cohesive energies (*E_c_*), bulk modulus (*B*_0_), and elastic constants (*C_ij_*) of hcp-Ti, fcc-Ti and bcc-Ti fitted from potential and obtained by experiments [49] or ab initio.

Physical Properties	hcp-Ti		fcc-Ti		bcc-Ti	
	Fitted	Experiments	Fitted	Ab Initio	Fitted	Ab Initio
*a* or *a*, *c* (Å)	2.836, 4.708	2.951, 4.684	4.008	4.097	3.217	3.246
*E_c_* (eV/atom)	4.789	4.850	4.790	4.792	4.769	4.738
*C*_11_ (Mbar)	1.553	1.624	1.177	1.400	0.962	0.986
*C*_12_ (Mbar)	0.887	0.920	1.094	0.972	1.126	1.150
*C*_44_ (Mbar)	0.392	0.467	0.530	0.592	0.580	0.454
*C*_13_ (Mbar)	0.745	0.690				
*C*_33_ (Mbar)	1.862	1.807				
*B*_0_ (Mbar)	1.080	1.051	1.122	1.115	1.071	1.096

**Table 3 materials-12-00432-t003:** The lattice constants (*a*, *c*), cohesive energies (*E_c_*), and bulk modulus (*B*_0_) of Ti-Nb hypothetical compounds fitted by the potential (first line) and calculated from ab initio (second line).

Physical Properties	Methods	TiNb_3_	TiNb	Ti_3_Nb
		L1_2_	B2	L1_2_
*a* or *a*, *c* (Å)	Fitted	4.313	3.394	4.235
	Ab initio	4.183	3.263	4.123
*E_c_* (eV/atom)	Fitted	6.696	6.124	5.447
	Ab initio	6.666	6.135	5.446
*B*_0_ (Mbar)	Fitted	1.174	1.506	1.244
	Ab initio	1.346	1.365	1.236

**Table 4 materials-12-00432-t004:** The lattice constants (*a*, *c*), cohesive energies (*E_c_*), and bulk modulus (*B*_0_) of Ti-Al hypothetical compounds fitted by the potential and calculated from ab initio (second line).

Physical Properties	Methods	AlTi_3_		AlTi	Al_3_Ti	
		L1_2_	D0_19_	B2	L1_2_	D0_22_
*a* or *a*, *c* (Å)	Fitted	4.047	5.691, 4.701	3.163	4.030	3.961, 8.590
	Ab initio	4.408	5.780, 4.647	3.186	3.983	3.854, 8.584
*E_c_* (eV/atom)	Fitted	4.764	4.716	4.397	4.159	4.105
	Ab initio	4.752	4.738	4.390	4.137	4.132
*B*_0_ (Mbar)	Fitted	1.222	1.114	1.093	1.090	0.970
	Ab initio	1.214	1.119	1.098	1.034	1.030

**Table 5 materials-12-00432-t005:** The heat of formation for each sub-binary system in Ti-Nb-Al ternary system [73].

Binary System	Enthalpy (KJ/mol)
Ti-Al	−92.23
Ti-Nb	1.30
Al-Nb	−9.99

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
