# Peer review of "Atomic-Approach to Predict the Energetically Favored Composition Region and to Characterize the Short-, Medium-, and Extended-Range Structures of the Ti-Nb-Al Ternary Metallic Glasses"

_materials, 2019, doi:10.3390/ma12030432_

Round 1

Reviewer 1 Report

Review of the article “Atomic-approach to predict the energetically favoured composition region and to characterize the short, medium and extended range structure of the Ti-Nb-Al ternary metallic glasses”.

In my opinion, the work is interesting and can give the community a better understanding of how the starting alloy composition affects the structural properties of Ti-Nb-Al alloys. The authors have carried out an good work based on the determination of the parameters that allow obtaining a greater capacity of formation of metallic glasses. The authors also have characterized the short medium and extended range structures depending on the composition for these alloys.

However, I should suggest:

a) The authors should revise the writing of the article. It is very difficult to read and understand. Some spelling mistakes are made, for example, in the use of verbal tenses. There are also some typographical errors. They should improve the presentation format within the article.

b) The authors should indicate the degree of the precision in the measurement of the lattice parameter and, also, of other calculated data. Or, where appropriate, the order of magnitude of the accuracy in which the parameters are determined.

c) It would be convenient to detail better in the article how the experimental value have been calculated and the production technique of these alloys. How does it affect in the results?

My conclusion is that this article could be published if the proposed modifications are carried out previously.

Author Response

Response to Reviewer 1 Comments

Point 1: The authors should revise the writing of the article. It is very difficult to read and understand. Some spelling mistakes are made, for example, in the use of verbal tenses. There are also some typographical errors. They should improve the presentation format within the article.

Response 1: Thanks for the referee’s advice. We have checked the errors of wording and grammar throughout the text and made corresponding corrections in the revised manuscript.

Point 2: The authors should indicate the degree of the precision in the measurement of the lattice parameter and, also, of other calculated data. Or, where appropriate, the order of magnitude of the accuracy in which the parameters are determined.

Response 2: The following settings ensure that the calculation meets the energy convergence and the accuracy requirements

In the present work, the first-principles calculations were carried out by the Cambridge Serial Total Energy Package (CASTEP) [1] of Materials Studio, in which the DFT and the Kohn-Sham approach was used to calculate the fundamental eigen values [2]. The interaction between valence electrons and core electrons was treated under the pseudo-potential approximation and the plane-wave approach [3]. The exchange-correlation energy was evaluated with the help of the Perdew-Wang functional (PW91) approach [4]. To reduce the basis set of plane wave functions used to describe the real electronic functions, ultra-soft pseudo-potentials [5] were implemented. The cutoff energy of the plane wave is 700 eV. A Monkhorst-Pack k-point grid separation of 0.02 Å1 was employed for the integration over the Brillouin zone according to the Monkhorst-Pack scheme [6]. The self consistent field (SCF) accuracy was set to 5×10-6 eV. After conducting the above settings, it was sufficient to reduce the error in the total energy to less than 1 meV/atom, and the convergence of the total force is less than -0.01 meV/A.

We have realized that this assertion is inaccurate, and the correction has been made which we hope meet with the reviewer’s approval:

In order to ensure that the calculations meet the requirements for energy convergence and accuracy. In the calculation process, the exchange correlation functional was constructed based on the generalized gradient approximation of Perdew and Wang (PW91) [51]. The Brillouin zones of different structures employed the same k-points grid separation of 0.02 Å1. The cutoff energy of the plane wave is 700 eV.

(Line 95-99, Page 3 in the manuscript)

REFERENCES

[1] Clark, S.J.; Segall, M.D.; Pickard, C.J.; Hasnip, P.J.; Probert, M.J.; Refson, K. First principles methods using CASTEP. Z. Krist. 2005, 220, 567-570.

[2] Kohn, W.; Sham, L.J. Self-consistent equations including exchange and correlation effects. Phys. Rev. 1965, 140, 1133-1138.

[3] Payne, M.C.; Teter, M.P.; Allan, D.C.; Arias, T.A.; Joannopoulos, J.D. Iterative minimization techniques for ab initio total-energy calculations: molecular dynamics and conjugate gradients. Rev. Mod. Phys. 1992, 64, 1045-1097.

[4] Perdew, J.P.; Chevary, J.A.; Vosko, S.H.; Jackson, K.A.; Pederson, M.R.; Singh, D.J. Atoms, molecules, solids, and surfaces: applications of the generalized gradient approximation for exchange and correlation. Phys. Rev. B. 1992, 46, 6671-6687.

[5] Vanderbilt, D. Soft self-consistent pseudopotentials in a generalized eigenvalue formalism. Phys. Rev. B. 1990, 41, 7892-7895.

[6] Monkhorst, H.J.; Pack, J.D. Special points for brillouin-zone integrations. Phys. Rev. B. 1976,13, 5188-5192.

Point 3: It would be convenient to detail better in the article how the experimental value have been calculated and the production technique of these alloys. How does it affect in the results?

Response 3: In this work, the experimental values were found in the relevant literature, rather than calculated. We have rephrased the language to make the meaning less confusing.

The GFR of a metal system may vary when using different glass-producing techniques. Take the Ni–Mo system as an example, because of the high melting points of the constituent metals Ni and Mo, liquid melt quenching has not been able to obtain any metallic glass in the system so far, yet ion beam mixing and mechanical alloying techniques have produced the Ni-Mo metallic glasses in a wide composition range [1-4]. To avoid confusion, it's propose to name the experimentally observed GFR as the nominal GFR, since they are closely related to the applied glass-producing techniques.

From a physical point of view, a metal system should have an intrinsic GFR, which is governed or determined by the internal characteristics of the system itself. The intrinsic GFR reflects the maximum possible composition range/region energetically favored for the metallic glass formation. In principle, the intrinsic GFR of a metal system should be larger than any nominal GFR observed in experiment. In addition, if a specific glass-producing technique is more powerful than other glass-producing techniques, then the nominal GFR obtained by this technique could be larger than those obtained by other techniques, and moreover, could be closer to the intrinsic GFR of the system than those observed by other techniques. In summary, the larger the intrinsic GFR, the larger the amorphization driving force, the larger the corresponding GFA, the easier the metallic glass forms.

The above discussion is not included in the text. Nevertheless, some statements have been rephrased or added so as to better explain our point:

In practice, researchers frequently use some technical parameters to describe the GFA of an alloy or to compare the GFAs of two alloys. However, the GFA of a metal system may vary while using different glass-producing techniques. Thus, many long-standing criteria for predicting the glass-forming ability (GFA) of alloys have been put forward [14-16], which are independent of the glass-producing techniques and facilitate the comparison of GFAs of different alloys. Such as, the reduced glass transition temperature Trg (=Tg/Tm) [14], the supercooled liquid region ΔTxg (=Tx-Tg) [15] and the new parameter γ=Tx/(Tg+Tm) [16], where Tg, Tx and Tm are the glass transition temperature, onset crystallisation temperature and melting temperature, respectively.

(Line 39-46, Page 1-2 in the manuscript)

REFERENCE

[1] B.X. Liu, W.L. Johnson, M.A. Nicolet, S.S. Lau, Nuclear Instruments & Methods in Physics Research 209 (1983) 229-234.

[2] Z.J. Zhang, B.X. Liu, Journal of Applied Physics 76 (1994) 3351-3356.

[3] G. Cocco, S. Enzo, N.T. Barrett, K.J. Roberts, Physical Review B 45 (1992) 7066-7076.

[4] Z.C. Li, B.X. Liu, Chinese Physics Letters 16 (1999) 667-669.

Reviewer 2 Report

In this work, atomistic simulations were used for prediction of the ability of Ti-Nb-Al solid solutions to form of metallic glasses. For this, new interatomic potential functions were parameterized using experimental data and simulations at the density functional theory level. Predictions of chemical compositions of solid solutions that are expected to form metallic glasses correlate with previous experimental observations. In addition, the atomic structure of the glass composition showing the highest (predicted) glass forming ability is analyzed in detail and discussed in context with glass formation mechanisms and mechanical properties.

The results are clearly presented and their discussion is convincing. However, I have several questions and comments that should be considered prior publication.

General comments

1. The numbering of the references is incorrect. For example, the references for eqns. 1-3 should be [45-47] and not [44-46] or for the LAMMPS package it should be [54] instead of [53].

2. Apart from numbering, the Materials Studio program suite and the LAMMPS package are not properly cited.

Section 2

3. Which program was used for fitting of the potential parameters and calculation of the elastic properties using the obtained interatomic potential?

4. Ll. 77-78: “… Nb-Nb, Al-Al and Nb-Al cross potential parameters have been fitted by the author before.” Please provide a corresponding reference.

5. Ll. 78-79: “The physical properties (the lattice constants, elastic modulus, elastic constants and cohesive energies) of hcp-Ti metal were obtained from the relevant literature …” Please add corresponding references.

6. Ll. 82: “… several intermetallic compounds with different structures and alloy compositions were selected …” Do you mean the compounds listed in Tables 3 and 4? How were these structures obtained (from literature or predicted in this work)?

7. Figure 1 compares results calculated with the new interatomic potentials with solid state equation of states. What was used for calculation of the Rose EOS, experimental or ab initio data (such as bulk moduli, etc.)?

8. More details on the DFT calculations are necessary (exchange-correlation functional, k-space sampling, cutoff for plane-wave basis set, etc.).

Section 3

9. The sentence in ll. 138-140 is somewhat misleading since a formation energy usually refers to a formation reaction. However, in this work the energy difference between crystalline and amorphous structure models with the same chemical composition was used for evaluation of the amorphous driving force (e.g., in Fig. 4), or not?

10. In l. 313, it is mentioned that the “open visualization tool” was used for the Voronoi tessellation analyses. It would be desirable to include this information to section 3.1 along with a corresponding reference.

Section 4

11. Including a brief explanation of the notation for Voronoi polyhedrons “” in the beginning of section 4.1. would facilitate reading for non-expert readers.

12. You used an equilibration employing the NPT ensemble at relatively low temperature (300 K) for generation of the glass structure models. However, simulated annealing (melt-and-quench) simulation procedures were frequently used in the literature for predictions of (low-energy) glass structures. Do you expect that both simulation procedures yield similar glass structure models?

13. In the discussion of Fig. 9 (ll. 306-326), the distribution of free volume is linked to the macroscopic mechanical properties. Does it mean that the Ti15Nb45Al40 glass shows anisotropic mechanical properties? Does it have any effect on the elastic properties (e.g., stiffness tensor)?

Author Response

Response to Reviewer 2 Comments

Point 1: The numbering of the references is incorrect. For example, the references for eqns. 1-3 should be [45-47] and not [44-46] or for the LAMMPS package it should be [54] instead of [53].

Response 1: According to the reviewer's suggestion, the numbering of the references in the manuscript has been checked and updated.

Point 2: Apart from numbering, the Materials Studio program suite and the LAMMPS package are not properly cited.

Response 2: The Materials Studio program suite and the LAMMPS package have been corrected and cited properly.

Point 3: Which program was used for fitting of the potential parameters and calculation of the elastic properties using the obtained interatomic potential?

Response 3: A MATLAB code developed by the researcher’s group was used for fitting of the potential parameters and calculation of the elastic properties using the obtained interatomic potential.

We have added this statement in the manuscript according to the reviewer:

In this work, we used a MATLAB code developed by the researcher’s group for the fitting of the potential parameters.

(Line 84-85, Page 3 in the manuscript)

Point 4: Ll. 77-78: “… Nb-Nb, Al-Al and Nb-Al cross potential parameters have been fitted by the author before.” Please provide a corresponding reference.

Response 4: We have added this literature in the manuscript according to the reviewer:

[48] Cai, B.; Yang, M. H.; Liu, J. B. Atomistic simulation study of favored compositions of Ni-Nb-Al metallic glasses. Sci. China. Tech. Sci. 2018, 61, 1829-1838.

Point 5: Ll. 78-79: “The physical properties (the lattice constants, elastic modulus, elastic constants and cohesive energies) of hcp-Ti metal were obtained from the relevant literature …” Please add corresponding references.

Response 5: In case that you didn’t notice, the corresponding reference has been marked on the Ll. 94 of the manuscript, next to Table 2. In order to make it easier to read, we also added a reference to the sentence:

The physical properties (the lattice constants, elastic modulus, elastic constants and cohesive energies) of hcp-Ti metal were obtained from the relevant literature [53].

(Line 87-89, Page 3 in the manuscript)

REFERENCE

[53] Sabeena, M.; Murugesan, S.; Anees, P. Crystal structure and bonding characteristics of transformation products of bcc beta in Ti-Mo alloys. J. Alloy. Compd. 2017, 705, 769-781.

Point 6: Ll. 82: “… several intermetallic compounds with different structures and alloy compositions were selected …” Do you mean the compounds listed in Tables 3 and 4? How were these structures obtained (from literature or predicted in this work)?

Response 6: Yes, the selected compounds were all listed in Tables 3 and 4. These structures were obtained from literature or calculated by CASTEP. For the sake of understanding, we have already clearly indicated in the table that a column is an experimental value or a calculated value.

Point 7: Figure 1 compares results calculated with the new interatomic potentials with solid state equation of states. What was used for calculation of the Rose EOS, experimental or ab initio data (such as bulk moduli, etc.)?

Response 7: In this work, experimental datas, such as bulk moduli, lattice constant, and cohesive energy were used for the calculation of the Rose EOS.

Point 8: More details on the DFT calculations are necessary (exchange-correlation functional, k-space sampling, cutoff for plane-wave basis set, etc.).

Response 8: More details about DFT calculations are added to the manuscript:

In order to ensure that the calculation meets the energy convergence and the accuracy requirements. In the calculation process, the exchange correlation functional was constructed based on the generalized gradient approximation of Perdew and Wang (PW91) [51]. The Brillouin zones of different structures employed the same k-points grid separation of 0.02 Å1. The cutoff energy of the plane wave is 700 eV.

(Line 95-99, Page 3 in the manuscript)

Point 9: The sentence in ll. 138-140 is somewhat misleading since a formation energy usually refers to a formation reaction. However, in this work the energy difference between crystalline and amorphous structure models with the same chemical composition was used for evaluation of the amorphous driving force (e.g., in Fig. 4), or not?

Response 9: We have realized that the statement of formation energy is improper, we have rephrased this sentence to make the words less confusing:

From a thermodynamics viewpoint, the energy difference between the amorphous phase and the idea solid solution, is the driving force for the solid solution to amorphize.

(Line 152-154 Page 5 in the manuscript)

Point 10: In l. 313, it is mentioned that the “open visualization tool” was used for the Voronoi tessellation analyses. It would be desirable to include this information to section 3.1 along with a corresponding reference.

Response 10: The “open visualization tool” mentioned in the manuscript was a software called “OVITO”. It has an Voronoi tessellation analysis modifier. This analysis modifier calculates the Voronoi tessellation of the simulation box, taking the particle positions as Voronoi cell centers. By default two quantities are output by the analysis modifier for each particle: The volume of the particle's Voronoi cell and the number of faces the Voronoi cell has.

We have rephrased this sentence to make the words less confusing:

The Voronoi tessellation of the simulation cell was calculated by the open visualization tool (OVITO) [77].

(Line 335-336 Page 12 in the manuscript)

Point 11: Including a brief explanation of the notation for Voronoi polyhedrons “1,n2,n3,n4>” in the beginning of section 4.1. would facilitate reading for non-expert readers.

Response 11:

According to the reviewer's suggestion, we have made the following related supplements:

According to the Voronoi tessellation analysis [60-62], an envelope of a family of perpendicular bisectors between a central atom and all of its nearby atoms constitutes the surface of a Voronoi polyhedron for the central atom, which can be differentiated by the indices3, n4, n5, n6,...>, where ni stands for the number of i-edged faces of the Voronoi polyhedron andis the total coordination numbers (CNs).

(Line 159-163 Page 5 in the manuscript)

REFERENCE

1. Finney, J.L. Random Packings and the Structure of Simple Liquids. I. The Geometry of Random Close Packing. Proc. Roy. Soc. Lond. A. 1970, 319, 479-493.

2. Hui, X.; Fang, H.Z.; Chen, G.L. Atomic structure of Zr41.2Ti13.8Cu12.5Ni10Be22.5 bulk metallic glass alloy. Acta. Mater. 2009, 57, 376-391.

3. Wang, Q.; Li, J.H.; Cui, Y.Y. Calculation of driving force and local order to predict the favored and optimized compositions for Mg-Cu-Ni metallic glass formation. J. Appl .Phys. 2013, 114, 153503.

Point 12: You used an equilibration employing the NPT ensemble at relatively low temperature (300 K) for generation of the glass structure models. However, simulated annealing (melt-and-quench) simulation procedures were frequently used in the literature for predictions of (low-energy) glass structures. Do you expect that both simulation procedures yield similar glass structure models?

Response 12: As we all know, metallic glasses have been obtained through a variety of producing techniques, such as solid-state amorphization (SSA), liquid melt quenching (LMQ), mechanical alloying (MA) and hydrogen-induced amorphization, and so on. A review of such experiments has been published [1]. The amorphous phase produced by solid-state-amorphization appears to be structurally very similar to that obtained by sputter deposition or melt quenching of the same alloy [2]. Other experiments have suggested that radiation-induced amorphization and melting of a crystalline compound are manifestations of the same first-order phase transition. Both exhibit features of a first-order phase transformation (e.g., nucleation and growth of the disorder phase) [3]. In addition, our recent published papers have demonstrated that there exists no pronounced differences in local atomic structure and mechanical behavior of metallic glasses between solid-state amorphization (SSA) and liquid melt quenching (LMQ) production methods [4]. In addition, solid-state-amorphization method was also frequently used in the literature for predictions of glass structures [5,6].

REFERENCE

[1] Russell, K.C. Phase-stability under irradiation. Prog. Mater. Sci. 1984, 28, 229-434.

[2] Beck, H. Hans-Joachim, Güntherodt. Glassy Metals III. Springer Berlin Heidelberg, 1994, 5-6.

[3] Johnson, W. L. Thermodynamic and kinetic aspects of the crystal to glass transformation in metallic materials. Progr. in Mater. Sci. 1986, 30, 81-134.

[4] Yang, M. H.; Li, J. H.; Liu, B. X. Proposed correlation of structure network inherited from producing techniques and deformation behavior for Ni-Ti-Mo metallic glasses via atomistic simulations. Sci. Rep. 2016, 6, 29722.

[5] Lin, C.; Yang, G. W.; Liu, B. X. Prediction of solid-state amorphization in binary metal systems. Phys. Rev. B. 2000, 61,15649-15652.

[6] Gong, H. R.; Kong, L. T.; Lai, W. S. Atomistic modeling of solid-state amorphization in an immiscible Cu-Ta system. Phys. Rev. B. 2002, 66,104204.

Point 13: In the discussion of Fig. 9 (ll. 306-326), the distribution of free volume is linked to the macroscopic mechanical properties. Does it mean that the Ti15Nb45Al40 glass shows anisotropic mechanical properties? Does it have any effect on the elastic properties (e.g., stiffness tensor)?

Response 13: Metallic glass does not behave as anisotropic [1]. Zhao Y. Y. and Louzguine-Luzgin D. V. et al. have investigated the correlation between free volume change (∆Vf) and elastic properties (e.g. notch toughness) of the metallic glass at different relaxed states [2, 3]. Their results demonstrate that the distribution of free volume has an effect on the elastic properties of metallic glass.

A brief discussion involving recent experimental observations is also added in Section 4.2:

Zhao Y. Y. and Louzguine-Luzgin D. V. et al. have investigated the structural relaxation and its influence on the mechanical properties and elastic properties of Cu-Zr-Ti-Pd and Mg-Zn-Ca bulk metallic glass (BMG) [75,76]. It’s found that the distribution of free volume has an effect on the elastic properties of metallic glass.

(Line 325-328, Page 12 in the manuscript)

[1] 汪卫华. 金属玻璃研究简史. 物理. 2011, 40 (11), 701-709.

[2] Zhao, Y. Y.; Zhao, X. Structural relaxation and its influence on the elastic properties and notch toughness of Mg-Zn-Ca bulk metallic glass. J. Alloy. Compd. 2012, 515, 154460.

[3] Louzguine-Luzgin, D. V.; Yavari, A. R.; Fukuhara, M. Free volume and elastic properties changes in Cu-Zr-Ti-Pd bulk glassy alloy on heating. J. Alloy. Compd. 2007, 431, 0-140.

Round 2

Reviewer 2 Report

The revised manuscript is considerably improved compared to its original version and suitable for publication in Materials.

Just two minor comments:

According to the LAMMPS homepage (https://lammps.sandia.gov/cite.html) the paper of Steve Plimpton (J Comp Phys, 117, 1-19) should be included in the reference of the LAMMPS package.

Regarding point 12: In my opinion, it would be desirable to briefly mention in the text that both atomistic simulation procedures (LMQ and SSA) yield similar atomic structures as shown in one previous publication of the authors (Sci. Rep. 2016, 6, 29722).

Author Response

Response to Reviewer 2 Comments

Point 1: According to the LAMMPS homepage (https://lammps.sandia.gov/cite.html) the paper of Steve Plimpton (J Comp Phys, 117, 1-19) should be included in the reference of the LAMMPS package.

Response 1: We have added this literature in the manuscript according to the reviewer:

It is possible to predict the GFR of the alloy system with the help of Large-scale Atomic/Molecular Massively Parallel Simulator (LAMMPS) packages [56, 57].

(Line 136-138, Page 5 in the manuscript)

REFERENCE

[57] Plimpton, S. Fast parallel algorithms for short-range molecular dynamics. J. Comput. Phys. 1995, 117, 1–19.

Point 2: Regarding point 12: In my opinion, it would be desirable to briefly mention in the text that both atomistic simulation procedures (LMQ and SSA) yield similar atomic structures as shown in one previous publication of the authors (Sci. Rep. 2016, 6, 29722).

Response 2: According to the reviewer's suggestion, we have made the following related supplements:

It is necessary to mention that different atomistic simulation procedures, such as liquid melt quenching (LMQ) and solid-state amorphization (SSA), yield similar atomic structures [21].

(Line 57-59, Page 2 in the manuscript)

REFERENCE

[21] Yang, M. H.; Li, J. H.; Liu, B. X. Proposed correlation of structure network inherited from producing techniques and deformation behavior for Ni-Ti-Mo metallic glasses via atomistic simulations. Sci. Rep. 2016, 6, 29722.